# Comparative Evaluation of the Tribological Properties of Polymer Materials with Similar Shore Hardness Working in Metal–Polymer Friction Systems

**DOI:** 10.3390/ma16020573

**Published:** 2023-01-06

**Authors:** Daniel Pieniak, Radosław Jedut, Leszek Gil, Włodzimierz Kupicz, Anna Borucka, Jarosław Selech, Grzegorz Bartnik, Krzysztof Przystupa, Zbigniew Krzysiak

**Affiliations:** 1Faculty of Transport and Computer Science, WSEI University, Projektowa 4, 20-209 Lublin, Poland; 2Tribology Center, Łukasiewicz Research Network-Institute for Sustainable Technologies (L-ITEE), ul. Pułaskiego 6/10, 26-600 Radom, Poland; 3Military Institute of Armoured and Automotive Technology, Okuniewska 1, 05-070 Sulejówek, Poland; 4Faculty of Security, Logistics and Management, Military University of Technology, Str. gen. S. Kaliskiego 2, 00-908 Warsaw, Poland; 5Faculty of Transport and Civil Engineering, Poznan University of Technology, Piotrowo 3, 60-965 Poznan, Poland; 6Institute of Mechanical Science, Vilnius Gediminas Technical University, J. Basanavičiaus Str. 28, LT-03224 Vilnius, Lithuania; 7Department of Automation, Lublin University of Technology, Nadbystrzycka 36, 20-618 Lublin, Poland; 8Faculty of Production Engineering, University of Life Sciences in Lublin, Głęboka 28, 20-612 Lublin, Poland

**Keywords:** polymer, friction, coefficient tests, roughness, Shore hardness, wear factor

## Abstract

This article presents comparative tests of contact strength and tribological wear resistance of polymer sliding materials of the polyamide group. The aim of this work was to study Shore hardness, indentation hardness, modulus, creep, relaxation, Martens hardness and sliding wear resistance of two commercial materials. One of these materials was produced with the recycling process in mind. Abrasion tests were performed against a stainless-steel ball (100CRr6) on a normal load of 5 N for 23,830 friction cycles. The samples were tested under dry friction conditions and taking into account the hydrothermal factor, the presence of which was assumed in the anticipated operating conditions. It was distilled water at a temperature of 50 °C. The volumetric wear of the samples under various environmental conditions was assessed and related to the mechanical properties, in particular, Shore hardness. This mechanical size, which characterizes the surface, was considered the most frequently used by engineers selecting polymeric materials for tribological applications in industry. The Shore hardness of both materials was similar, which may indicate similar tribological performance properties. However, research and analysis indicate the need to use measures that directly correspond to tribological wear. The friction and wear of both materials varied. The coefficient of friction in hydrothermal conditions was lower and the wear was higher than in the dry friction test. It seems that it was not hardness that determined the suitability in the anticipated operating conditions, but the ability to form a sliding layer on the friction surface. The properties of the material that has been envisaged as a replacement may be appropriate for the intended uses.

## 1. Introduction

Polymer plastics (also called plastics) are organic compounds obtained through chemical processes [1]. Mechanical engineering typically uses thermoplastic polymers (e.g., polyethylene, polyamide), resins and thermosetting polymers—duroplastics (e.g., epoxy and polyester resins), elastomers (e.g., polyisoprene, polybutadiene), and natural polymers (e.g., compressed wood, cellulose fibers) [2]. Thermoplastics and duroplastics form a group of plastomers. These are materials that are characterized by small deformations < 1% and a high Young modulus under low loads, while the upper limit of use is established by the glass transition temperature. Thermoplastics transform into a plastic state and harden when cooled. These materials are hot-formed and can be repeatedly recycled. These plastics have an amorphous structure; in the case of some plastics (e.g., PA) small crystalline areas are formed. Weak bonds melt easily, allowing them to form and hold a new shape when cooled. ABS is an example of this type of plastic. Duroplastics are made up of cross-linked macromolecules and have a thermally irreversible structure. The structure contains C-C covalent bonds much stronger than hydrogen bonds. These plastics are more cross-linked. The crosslinking is not broken when heated, so once cross-linked, the material cannot be heat-molded again. They are usually insoluble, but they swell [3]. Elastomers—at low stresses—show high deformation (up to 1000%), since their glass transition temperature is below room temperature, so the temperature range of use is highly elastic [4]. Today, polymeric materials play a significant role in mechanical engineering and transportation. They are used as volumetric elements in the design of kinematic machine nodes, where there is an interaction between the machine parts in surface contact [5,6]. Examples include plain bearings and gears. From an application point of view, the production of polymeric anti-friction coatings on the surfaces of metal components is also important [7]. There is frictional contact, and the interaction of elements is characterized by the friction coefficient. It is a dimensionless quantity that describes the relationship between friction force and load. Two types of friction coefficient can be distinguished: one that represents friction opposing the start of relative motion (approaching motion) and one that represents friction opposing the continuation of relative motion after the start of motion. The first is called the static friction coefficient and the second is called the kinetic friction coefficient [8]. Owing to the good to very good tribological properties of many polymer materials, they can be used in nodes where dry sliding friction is present [9]. This allows polymers to be used in a range of machine kinematic nodes, which is also enhanced by the polymers’ ability to self-lubricate while sliding on metal surfaces. A low friction coefficient combined with reasonable wear resistance can be achieved in polymer-metal sliding pairs without the use of an external lubricant. This is a definite advantage of polymers as sliding materials, especially in situations where the supply of lubricant to the friction node is structurally difficult [10]. Based on the state of the art, the authors of this paper carried out studies under dry sliding friction conditions, or, more accurately, as described in [11], technically dry for a metal-polymer association. A polyamide group material called Nylon was used for this purpose. There is a lack of complete knowledge that covers the performance of these materials relevant to applications in friction nodes. Application experiments in this area are conducted primarily in company laboratories and in industrial companies that are reluctant to share such knowledge. Furthermore, operational experience is derived from engineering practice and is often not provided under unstable and variable real-world operating conditions. As a result, information about the material in question is scarce. The only known information about the polyamide (Nylon) being studied is its hardness (on the Shore scale) [12], which is comparable to that of TECAMID 6 [13] used in machine kinematic nodes, which was chosen as a reference in the present study.

In response to the comments above, the authors presented a study that addresses the aforementioned problems. First, a comprehensive analysis of the compared materials was made, providing knowledge not only of hardness but also of surface modulus, surface creep resistance, surface relaxation, universal hardness, and wear resistance. These are key parameters for the proper selection of the material in terms of its applications. The tests were carried out in a standardized and stable laboratory environment. Furthermore, it was taken into account that this type of material is used in a highly corrosive environment, so the second stage of consideration and research concerned the application of polyamides to kinematic nodes exposed to operation in a humid environment with elevated temperatures. Water absorption can affect the mechanical properties of polymers and polymer composites [14]. Despite this, polymers are used in applications where contact with water is unavoidable and sometimes water is a lubricant [15]. In addition, the action of tensile thermal stresses causes microdamage to occur, which facilitates the penetration of water into the material. It is possible that water acts as a plasticizer within the structure of the polymer material, and its action leads to stress relaxation and reduction in stiffness and at the same time to the destruction of the polymer network. The mechanism of hydrothermal degradation in friction nodes is not yet recognized and requires additional research. Many manufacturers of sliding plastics state that, in engineering practice, plastic materials exposed to contact loads are selected based on Shore hardness, which is one of the mechanical parameters provided by the manufacturer.

Accordingly, the utilitarian purpose of the study was to comparatively evaluate the behavior of engineering polyamides of similar nominal hardness under frictional interaction with a metal component in concentrated contact. It was assumed that the recycled, ecologically obtained material proposed in the study could be a good alternative to many sliding polymers. The article consists of an introduction to the issue under study in which a literature review was conducted, the existing research gap was identified and the research undertaken was justified. The purpose of the study and the research hypothesis are also presented. The methods and materials used in the study are then presented (Section 2). Section 3 presents the results of the tests conducted. Section 4 analyzes and discusses the results obtained. It concludes by synthesizing the key findings from the study.

## 2. Material and Methods

### 2.1. Materials

In our own tests, samples of two polyamide polymer plastics used in machine kinematic nodes were used. These were produced by two different companies that specialize in plastics. The first material was polyamide 6 with the trade name TECAMID 6 natural, manufactured by Ensinger [13]. As it was delivered, it was an extruded rod with a diameter of 30 mm. The second material that was treated as a replacement for TECAMID 6 was a polyamide group material called Nylon, manufactured by Jiujiang Autai Rubber and Plastic Co., Ltd. in Jiujiang, China [12]. As delivered, it was a 30 mm diameter rod with the name “Nylon Rod Recycled” on it, from which we can infer that it is recycled plastic. Some performance characteristics of the materials are given in Table 1. Two types of specimens were needed for the study. For Shore hardness testing, five specimens of each material with a thickness of 8 mm were cut on a lathe. The samples were taken from bars with a diameter of 30 mm. For friction testing, 3 specimens of each material were made, their thickness was 3 mm, and their diameter was reduced to 29 mm to fit the tribometer holder. The same samples were used in microindentation tests. ATM’s Saphir 330 laboratory instrument was used to prepare the surface of the specimens. It is a double row grinder and a polisher with a working wheel diameter of 200–250 mm. The samples were ground with water with discs of decreasing granularity and then polished with a cloth with water.

### 2.2. Surface Roughness Tests

Roughness testing was performed on surfaces machined with a grinder-polisher. A VeecoDektak 150 contact profilometer was used. This profilometer allows for 2D topography and 3D surface measurements with a resolution of 0.01 µm in the *Z*-axis. The resolution of the measurement was specified as 0.1 µm. The analyses were the height of the highest elevation of the roughness profile (Rp), the depth of the lowest depression of the roughness profile (Rv), the highest height of the roughness profile (Rz), the arithmetic mean of the profile ordinates (Ra), the total height of the profile (Rt), and the mean square deviations of the profile (Rq). In addition, an analysis of the cross-sectional areas of the friction surfaces was performed, this quantity being designated S_AR_. Measurements of half-cross-sectional areas were made in ten sections along the perimeter of the friction track. The resolution of the measurement was specified as 0.1 µm. The value of the measurement path was in the range of 500 to 2000 μm (Figure 1), depending on the width of the flange.

### 2.3. Shore Hardness Measurement Method

The hardness test was carried out using a Shore “D” scale apparatus, designed to test hard and very hard plastics according to standard [16]. The camera scale ranges from 0 to 100 Shore degrees. The indenter test load in the D Shore method was 44,450 mN (5000 g). A Shore HPE II Bareiss durometer mounted on a BS 61 II Bareiss tripod was used in the study.

### 2.4. Indentation Hardness Tests

Indentation tests were performed on an Anton Paar MCT platform. The test method used is in accordance with the technical standard [17]. A Vickers indenter was used and the waveforms of force variation and indenter displacement deep into the surface of the specimens were recorded, according to the scheme presented in Figure 2. Measurements were carried out at a maximum loading force of 1 N and a loading/unloading speed of 500 mN/min. Mechanical and elastic parameters were determined. Indentation hardness *H_IT_* (1) was determined by the ratio of the highest normal force loading of the indenter *P*_max_ to the contact area of the indenter under maximum load *A*, according to the following formula [18]:(1)$$H_{IT}=\frac{P_{max}}{A}.$$

The converted Vickers hardness was obtained on the basis of relation (2) [18]:*HV_IT_* ≈ *H_IT_*/*10.58.*(2)

To calculate the elastic modulus of the *E_IT_* surface, the stiffness was determined from relation (3) [18]:(3)$$S=\frac{dP}{dh}=\beta \cdot (2/\sqrt{\pi })E^{*}\cdot \sqrt{A}.$$

The relation $\frac{dP}{dh}$ was determined from the indenter force-displacement diagram (Figure 2). In Equation (3), the parameter *β*, for any symmetric indenter, is taken equal to 1. For the Vickers indenter, the adjusted value of the parameter *β* = 1.0055 (Oliver 2004 [18]). The value *A* is a function of the depth *h_c_* and is determined from relation (4), according to [19]:(4)$$A=F\left(h_{c}\right)=24,504h_{c}^{2}+C_{1}h_{c}^{1}+C_{2}h_{c}^{\frac{1}{2}}+C_{3}h_{c}^{\frac{1}{4}}+C_{4}h_{c}^{\frac{1}{8}}+\ldots +C_{n}h_{c}^{\frac{1}{2n}}.$$

In Equation (4), the calculations are based on the constant *C_n_*, which expresses the geometry of the indenter. The method of determining the constant *C_n_* is described in paper [20]. In stiffness calculations, there is a surface layer modulus denoted by the letter *E*. This quantity is defined by Equation (5):(5)$$\frac{1}{E^{*}}=\frac{1-\nu^{2}}{E}+\frac{1-\nu_{i}^{2}}{E_{i}},$$
where:$\nu_{i}$—indenter Poisson ratio;$E_{i}$—indenter modulus.


Martens hardness is determined under an applied loading force and includes plastic and elastic deformation. Martens hardness is defined as the quotient of the loading force *P* and the indenter area *As(h)* at depth *h* [21]. For the Vickers indenter, Martens hardness is determined from the formula *HM = P/26.43* · *h2* (6) (parameter *h* as indicated in Figure 2).

The test further determined the indentation creep *C_IT_*. According to the definition, the *C_IT_* parameter is the change in indentation depth at a constant load expressed as percentage and was calculated based on the formula:(6)$$C_{IT}=\frac{h_{2}-h_{1}}{h_{1}}\cdot 100,$$
where *h*_1_ is the depth of penetration at time *t*_1_ after reaching force *F* (which is held constant), and *h*_2_ is the depth of penetration after time *t*_2_ of holding force *F* constant (Figure 3).
(7)$$R_{IT}=\frac{F_{1}-F_{2}}{F_{1}}\cdot 100$$

In the equation for the relaxation parameter, *F*_1_ is the force the assumed value of which was achieved at the corresponding penetration depth. The indenter was then kept at the same depth for a presumed time from *t*_1_ to *t*_2_ (the time is *t*_2_ − *t*_1_) until the force *F*_2_ was reached (Figure 4).

Analysis of the force-penetration depth curves (Figure 2) makes it possible to determine one more parameter. It is the ratio of the work of force on elastic *(W_e_*) to total *(W_total_*) deformation [23]:(8)$$\eta_{IT}=\left(\frac{W_{e}}{W_{total}}\right)\cdot 100.$$

### 2.5. Friction and Wear Tests

Friction tests were carried out for a metal-polymer association in a ball (made of 100Cr6 steel)—disc (the polymer under test) arrangement, i.e., in a concentrated contact. Tests under dry sliding friction conditions were performed on an Anton Paar THT tribometer (Figure 5). Liquid tests were performed on a CSM tribometer equipped with a measuring vessel and a liquid thermal conditioning system. Deionized water was used. The disc (specimen) and the ball (static partner) were completely immersed in water. The test temperature was set at 50 °C. The friction test parameters are presented in Table 2. Based on the measured value of wear expressed as furrow cross-sectional area ratio (*S_AR_*), volumetric wear was calculated according to the following equation:(9)$$V_{disc}=2\pi R\cdot S_{AR},$$
where:*V_disc_*—voluminal wear (μm^3^),*R*—radius of friction (7 mm = 7 × 10^3^ μm),*S_AR_ (Mean)*—mean cross-sectional area of the friction track (μm^2^).

## 3. Results

### 3.1. Roughness Test Results

The values of the model parameters are presented in Table 3. The *R_a_* parameter of the tested specimens varied slightly. Nylon showed a higher roughness considering all specimens and all measured parameters.

### 3.2. Shore Hardness Test Results

Figure 6 shows the statistical distributions (distribution series) of Shore hardness and descriptive statistics. The number of measurements (N), mean value (Mean), standard deviation (StdDv), maximum (Max), and minimum (Min) values were taken into account. The hardness of the materials was similar, but in the case of Tecamid 6 it was 8.16% higher. In addition, the statistical scatter of Nylon’s results was significantly higher. The results of the tests appear to be correct, as similar values were obtained and presented by [24].

### 3.3. Microindentation Test Results

Figure 7 shows the normal force-penetration depth curves. These are averaged curves. The shape of the curves is similar. A higher average penetration depth under 1 N load was achieved with Tecamid 6, but the residual (durable) penetration depth of this material was slightly lower than that of Nylon. This means that the share of elastic deformation was higher.

Figure 8 presents box plots of parameters obtained in the indentation test. The materials differed in hardness. An inverse relationship with the Shore hardness obtained in the test was shown. Nylon was characterized by a higher Vickers indentation hardness by 11.86%, a statistically significant difference, which was demonstrated in the statistical non-parametric Wilcoxon test (*p* = 0.002874) and the so-called Martens universal hardness by 16% (*p* = 0.002218). The parameters reduced modulus of elasticity and stiffness are also differentiated (by 30.19% and 20.58%, respectively), statistically significantly to the disadvantage of the Tecamid 6 material (*p* = 0.002218 for both parameters). For the parameters characterized by creep and relaxation (Figure 8), the differences in percentage terms were much lower, at 3.18% (*p* = 0.028057) and 5.7% (*p* = 0.410118), respectively. Slightly higher absolute values were obtained for Tecamid 6. But the differences were small and, in the case of the R_IT_ parameter, not statistically significant. This means that Tecamid 6 can be considered to have a slightly higher susceptibility to creep. In contrast, the parameter describing the contribution of elastic work to the deformation of the surface under the indenter showed higher differences (20.46% and *p* = 0.002218).

### 3.4. Results of Friction Coefficient Tests

Table 4 presents statistics of friction coefficients depending on the material and the specimens. The initial value was measured when the movement began. It varied the most, but in all cases of dry sliding friction it was <0.1, which can be considered a low friction coefficient. The average values obtained under dry sliding friction conditions for Nylon were similar, but the variation in the average value was higher for Tecamid 6, which could be caused by the unstable nature of friction. The plots show a higher variation in the friction coefficient for Tecamid 6, both during dry sliding friction and in heated liquid. In the case of a representative specimen of this material during dry sliding friction, there was a clearly staged course of friction coefficient variation, and the staging was also noticeable in friction involving a hydrothermal agent (Figure 9). In the first stage of dry sliding friction, up to about 9000 cycles, the friction was stable, then the friction coefficient increased intensively up to about 13,000 cycles. After that, there was a significant decrease up to about 16,000 cycles and then an increase and large fluctuations in the friction coefficient were seen, which can be associated with a significant deterioration of sliding properties. Clear peaks can be seen, with interpeak values ranging from 0.2 to 0.5. In friction tests with the hydrothermal agent, the variability was lower.

### 3.5. Results of Wear Tests

Figure 10 shows selected profilograms of furrow cross-sections formed on friction surfaces. Table 5 summarizes the results of measurements of the cross-sectional areas of the friction track furrow. The average values of the cross-sectional areas of the furrows vary. Higher values were measured on the friction surfaces of Tecamid 6 samples.

## 4. Analysis and Discussion

Traditionally, tribology testing has focused on reliability, ensuring safe and continuous operation of machine components. Today, due to the increased pressure to reduce energy consumption, efficiency is becoming increasingly important [25]. Meeting these criteria requires the use of materials to their fullest extent, and this is one of the reasons for seeking a reduction in material consumption and using recycled plastics. In addition, reducing friction resistance has a positive effect on energy consumption. Engineers can use diverse criteria when selecting materials, and they use mainly such plastics as thermoplastics and duroplastics. Importantly, for material efficiency, these plastics have favorable specific strength compared to other structural materials. Additionally, the cost of obtaining the material is usually more favorable compared to the cost of metals. Therefore, polymeric engineering plastics are increasingly used in mechanical engineering [26], especially mass-produced ones. Also relevant to the use of polymers as sliding materials may be the current scientific developments based on social considerations, which have led to the formulation of the principles of green tribology, including minimization of heat and energy dissipation, reducing the emission of heat to the environment and optimization of this phenomenon by various methods; minimization of wear and tear, which is closely related to green tribology, since wear and tear leads to a shorter life of machine components/elements, which requires recycling/replacement, which in turn leads to environmental degradation through, among others, excessive waste; reduction or complete elimination of lubrication replaced by self-lubrication; natural lubrication—natural oils—can be used as they are environmentally friendly, but also are a key food ingredients in developing countries; biodegradable lubrication; principles of sustainable chemistry and green engineering; and biomimetic approaches, mimicking nature/natural systems as they are more environmentally friendly [27]. The good sliding properties of polymers, self-lubricating ability, potential for weight reduction, miniaturization of friction system components, less noise from mating components, and partial recyclability, as in the case of one of the materials tested, fit in with some of the principles of green tribology. Therefore, attempts to expand applications and search for suitable polymeric materials should be carried out, although the range of applications is limited, since plastics have low thermal resistance (possibility of thermal degradation) and high coefficients of thermal expansion (the need for large clearances) [7]. One polymeric material of key importance in technology is polyamide, not only because of its performance characteristics but also because of the large number of plants producing such plastics, which translates into availability and competitive pricing. Parts made of this material are widely used in mechanical engineering and in the automotive industry [28], and these include bushings and plain bearings, friction inserts, support and guide wheels, conveyor rollers, manipulator pulleys, wheel and roller bushings, pulleys and pulley liners (coatings), cams, spring washers, hammers, scrapers, sprockets, sprocket teeth, sealing rings, pull bolts, star crosses, cutting and chopping plates, insulators, and others. Our own tests were focused on applications involving dry sliding friction, hydrothermal friction, and cooperation with metal components. This is a common application of polymer plastics [29]. In applications where there is friction between a polymer component and a much harder metal component, the hardness of the polymer plays an important role [30,31]. Traditional strategies for improving surface-wear resistance have focused on modifying the surface with various treatments to improve surface hardness [32]. In the plastics industry, Shore hardness measurement is commonly used in assessing surface strength [33]. The average Shore hardness of the tested materials differed by 8%, with Tecamid 6 showing a higher hardness. In an engineering evaluation, such a difference can be considered insignificant, and the two materials tested can be considered similar because of their surface resistance to mechanical damage. However, despite similar Shore hardness, the test results indicate that the sliding properties and wear resistance of the tested materials vary, also due to the hydrothermal factor of the operating environment. More reproducible sliding properties were obtained with the recycled Nylon material, which seems surprising. This behavior may be related to the deposition of polymer-disc wear products on the friction surface of the metal counter specimen. The articles published so far [34,35,36] also indicate that, as a result of long-term tribological processes, the polymer–metal friction pair becomes a polymer–polymer cooperation, which means that this behavior of the polymer material in the friction pair is expected. In the case of Tecamid 6, the increase in the friction coefficient and fluctuation of the same after exceeding about 10,000 dry sliding friction cycles and 18,000 friction cycles with heated fluid (depending on the specimen) and the uniqueness of the sliding properties may indicate that this material does not behave predictably, and it will be difficult to assess the durability of the application in the sliding node, and if so, then only in the low-cycle range. According to [37], this behavior depends on the viscoelastic properties of the material and is related to creep susceptibility and stress relaxation. The indentation test showed that these parameters are higher for Tecamid 6. Furthermore, the authors of the paper [37] explain that the frictional force can increase when a particle of the wear product of a viscoelastic material or polymer adheres to the friction surface and is further deformed, resulting in additional energy consumption to overcome frictional resistance. It is possible that this phenomenon caused the marked change in the friction coefficient of Tecamid 6. The synergistic interaction of contact forces and the hydrothermal factor of the operating environment caused large changes in friction and wear of the materials tested. As reported in Gebretsadik (2020) [14], the friction and wear of polymers depend on the environment, regardless of whether they operate in dry or wet conditions. In our own tests under hydrothermal conditions, the friction coefficient of most specimens was lower and wear was many times higher compared to that under dry sliding friction conditions. In paper [14] it was shown that for distributed contact, linear wear in deionized water is approximately 10 times higher compared to dry sliding friction conditions. In the tests presented in this paper, where elevated operating temperature was taken into account in addition to moisture effects and frictional contact was concentrated, the differences were smaller, but expressed as volumetric wear rather than linear. Furthermore, higher differences in wear under dry and hydrothermal friction conditions were shown for Tecamid 6, and lower for Nylon. This means that hydrothermal factors, which are predicted to occur in the operating environment, have a significant impact on the tribological resistance of the surface. In this situation, it seems that an engineering approach in material selection based solely on the Shore hardness of the material may not be sufficient in assessing the suitability of a material in a sliding node. This is not to say that the problem described cannot have a practical solution for the designer. For this purpose, calculations were made of the widely used wear index and Archard’s coefficient, which is practically constant, below the limit of the product pv (p—pressure, v—sliding velocity) [37]. The values of these measures make it possible to compare the tribological properties of the tested materials with those obtained in tests conducted in other laboratories. The calculation used the formula presented in the article [38]:(10)$$k=\frac{V_{discs}}{F_{N}\cdot L},\;$$where:*V_discs_*—volume wear (mm^3^);*F_N_*—normal force—the load in the friction test (N);*L*—friction path [m].

The calculation results are presented in Table 6.

The values of the wear index *k* quantitatively confirm the better anti-wear properties of Nylon. Another formula frequently used in benchmarking is Archard’s coefficient. It was calculated according to the following formula presented in paper [39]:(11)$$K=\frac{H\cdot V}{F_{N}\cdot L},$$
where:*H*—Vickers hardness of the less hard material of the given pair [MPa];*V*—volume wear [mm^3^];*F_N_*—normal force [N];*L*—friction path [mm].

A lower value of the Archard coefficient is favorable. When applying this factor to the results of the dry sliding friction test, the differences between sliding plastics were much more pronounced and the differences between material specimens were much lower. Unfortunately, Archard’s coefficient varied much more when applied to the results of friction tests with a hydrothermal agent, which may indicate a much higher complexity of friction and wear under hydrothermal conditions. For such applications, experimental tribological tests should be performed that mimic operating conditions to a possibly higher degree, but unfortunately they are time-consuming and expensive.

Tribological wear is also affected by the elastic modulus of the surface. Currently, polyamide is used to manufacture, among other things, gears, which are used in gear trains loaded with low to medium torque [40]. In the paper [41] it was found that due to their low elastic modulus, polymer gears deform, and their profile outline wears out. In the test results presented, Nylon showed a higher modulus of elasticity, which was also translated into the tribological test results. Other researchers [42] use the H/E ratio (indentation hardness HV_IT_ to the reduced surface elastic modulus E* based on [18,19,20]), determined by the Oliver-Pharr method [43], to assess suitability in sliding nodes. This coefficient combines the ability to achieve the greatest possible elastic deformation (low modulus) and the ability to minimize permanent deformation (high hardness). The results of the analysis of this relationship are shown in Figure 11 (they were obtained in a test carried out according to the method described in point 2.4 of this work). A higher ratio value was obtained for Tecamid 6. However, as shown earlier, this does not translate into material behavior in friction tests. This confirms the need for friction testing and/or the use of other coefficients.

## 5. Conclusions

Based on research and analysis, the main conclusions were formulated.
The use of polymeric materials in engineering applications is growing every year. Many polymeric materials, including polyamides, are reported to be tough and wear resistant. The polyamides used in our own tests were used in nonlubricated metal–polymer friction nodes where contact stresses occur. According to the specialized literature [44], hardness is particularly important for structures with contact stresses. It has also been shown [45] that the hardness of many polymer plastics is correlated with the friction coefficient. The Shore hardness of the two materials is similar, which may indicate a similarity in tribological performance. However, tests and analyses indicate the need for measures that directly correlate with tribological wear.In our own tests, the friction and wear of the two plastics varied. It seems that it was not the hardness that determined the tribological properties, but rather the ability to form a sliding layer. In papers [46,47] it was noted that in polymers that contain a so-called “slip agent,” it tends to migrate to the surface of the polymeric component and form a waxy layer that lowers the friction coefficient and stabilizes friction. Presumably self-lubricating nylon is better. It is possible that the stable friction pattern and the lower wear of the Nylon material are decisive for the sliding properties. Macrodeformation of the surface of a polymeric component in the friction process in concentrated contact can be of crucial importance. It depends, among other things, on the elastic modulus, which was higher for Nylon, and on the resistance to repeated deformation (24,000 friction cycles were performed). Macrodeformation occurs mainly in the form of a bulge of less hard material (polymer) in front of the face of the friction element (metal ball). This behavior affects the energy processes at the friction–pair interface, where part of the mechanical energy is converted into thermal energy [48]. An increase in temperature at the friction interface increases the deformability of the polymer, and at the same time, leads to an intensification of macrodeformation and frictional resistance. It is possible that this mechanism caused the sliding properties of Tecamid 6 to deteriorate after about 10,000 dry sliding friction cycles. Furthermore, friction coefficients under hydro-thermal test conditions at 50 °C were not higher. It seems that the elastohydrodynamic effect that occurs at the interface of concentrated lubricated friction nodes may have an influence.

## Figures and Tables

**Figure 1 materials-16-00573-f001:**
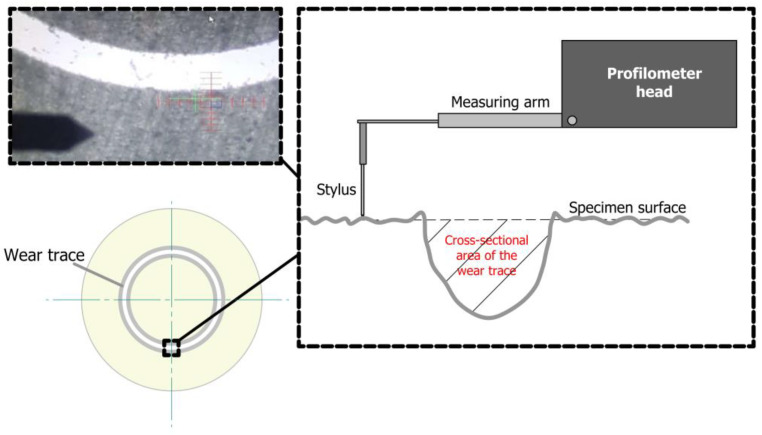
Measurement of the cross-sectional area of the wear trace of PA6 specimen on a contact (stylus) profilometer.

**Figure 2 materials-16-00573-f002:**
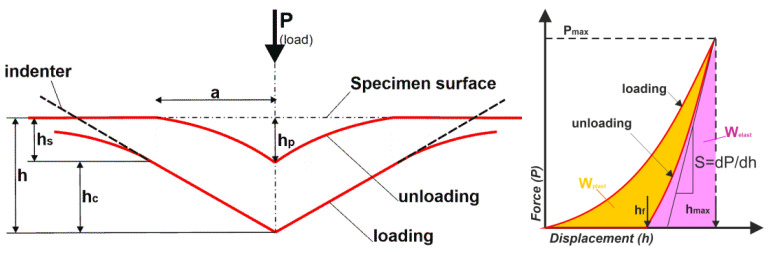
Schematic of the indenter contact with the test surface and theoretical force-displacement characteristics with important test parameters marked [22].

**Figure 3 materials-16-00573-f003:**
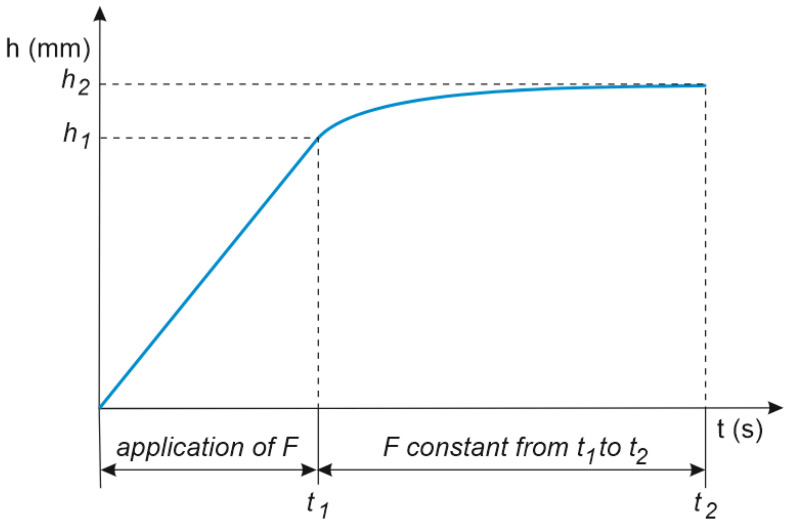
C_IT_ creep parameter determination method.

**Figure 4 materials-16-00573-f004:**
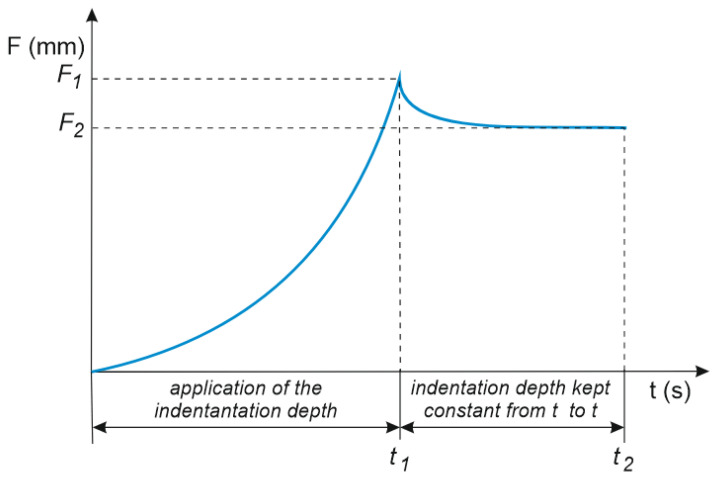
R_IT_ relaxation-parameter determination method.

**Figure 5 materials-16-00573-f005:**
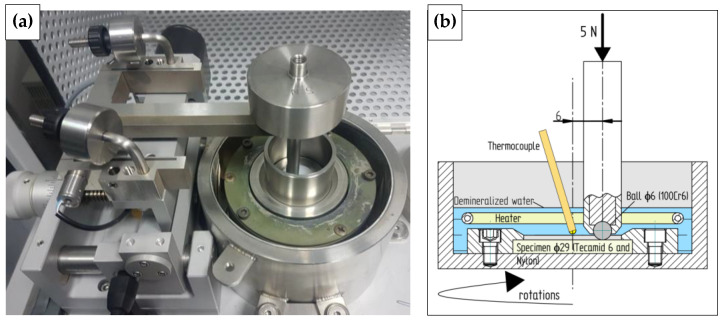
THT tribometer test setup (dry sliding friction) (**a**) and CSM tribometer test scheme (friction in demineralized water at 50 °C (**b**).

**Figure 6 materials-16-00573-f006:**
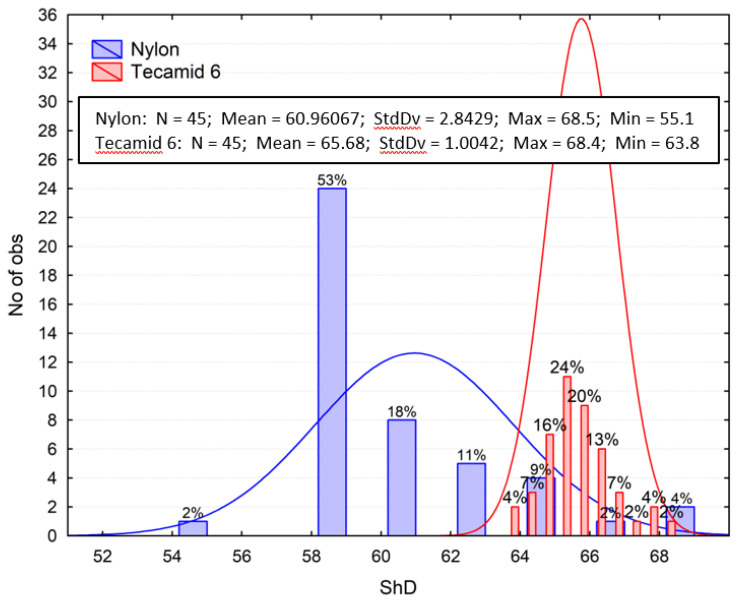
Distribution series of Shore hardness test results.

**Figure 7 materials-16-00573-f007:**
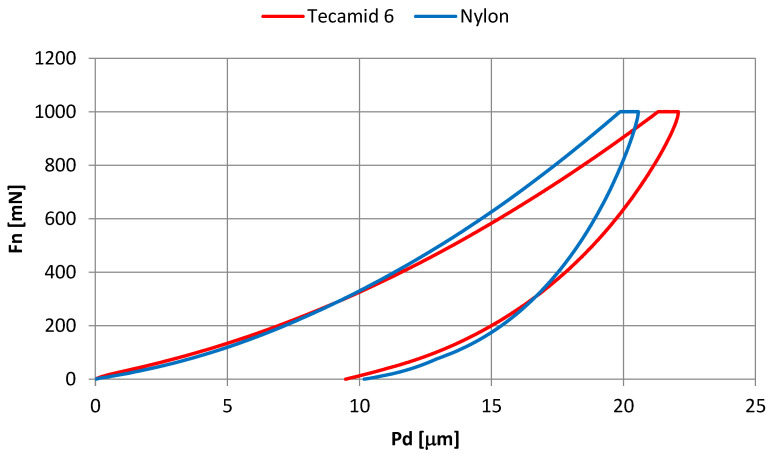
Mean normal curves (Fn)—penetration depth of the indenter (Pd).

**Figure 8 materials-16-00573-f008:**
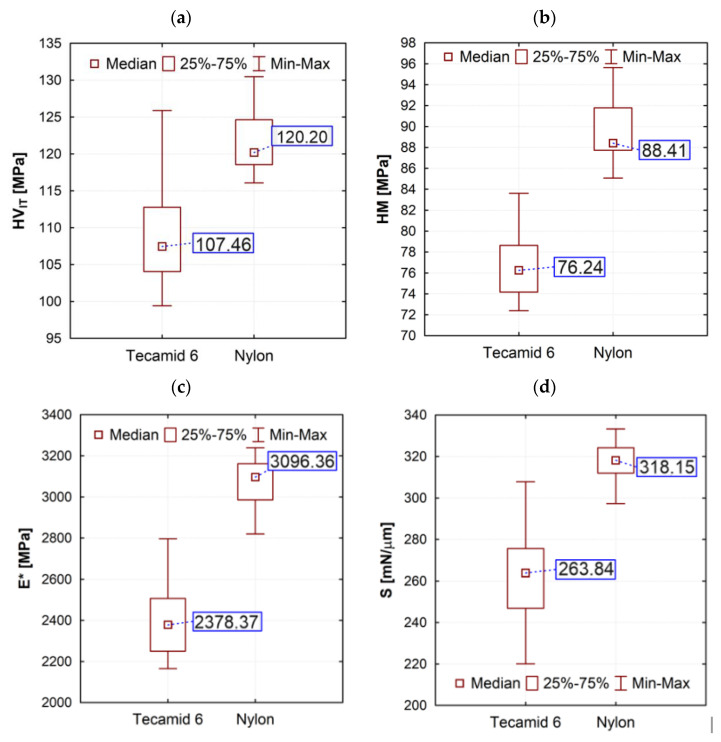
Box plots of the parameters obtained in the indentation test: (**a**) Vickers indentation hardness; (**b**) Martens hardness; (**c**) indentation modulus; (**d**) indentation stiffness; (**e**) indentation creep; (**f**) indentation relaxation; (**g**) indentation ratio of the work of force on elastic to total deformation.

**Figure 9 materials-16-00573-f009:**
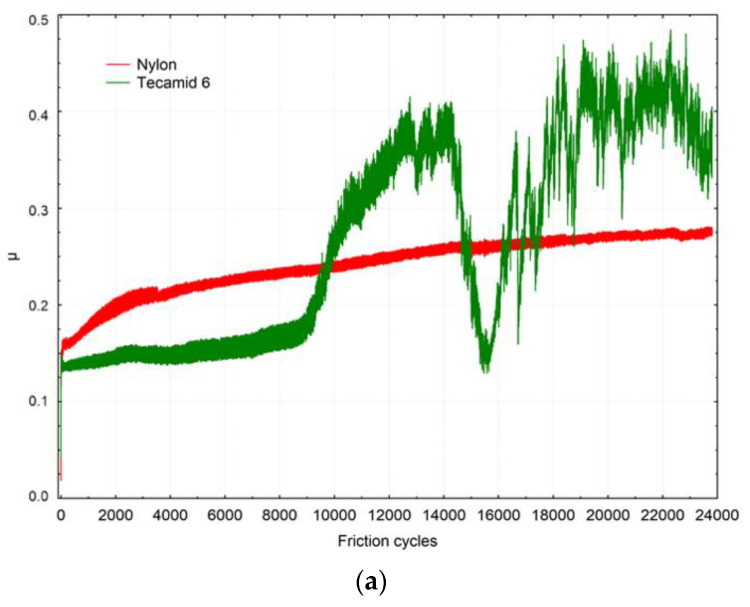
Comparison of friction coefficient results depending on the number of friction cycles and materials tested: (**a**) dry test; (**b**) wet test 50.

**Figure 10 materials-16-00573-f010:**
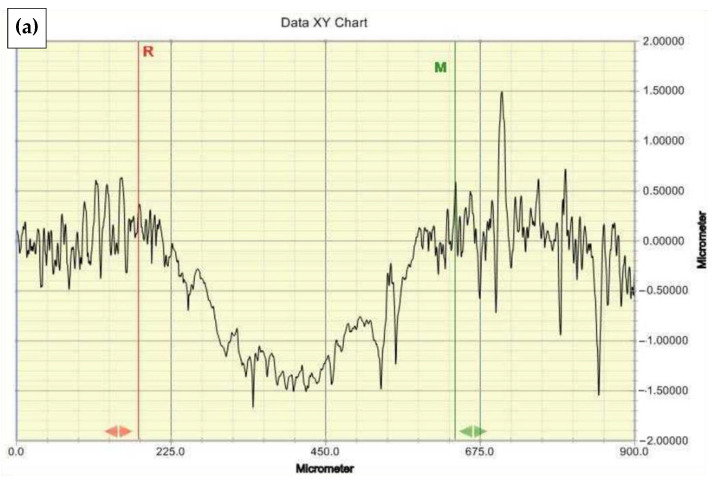
Selected cross-sections of surface wear profiles of Nylon and Tecamid 6 specimens: (**a**) Nylon dry sliding; (**b**) Nylon wet sliding 50 °C; (**c**) Tecamid 6 dry sliding; (**d**) Tecamid wet sliding 50 °C.

**Figure 11 materials-16-00573-f011:**
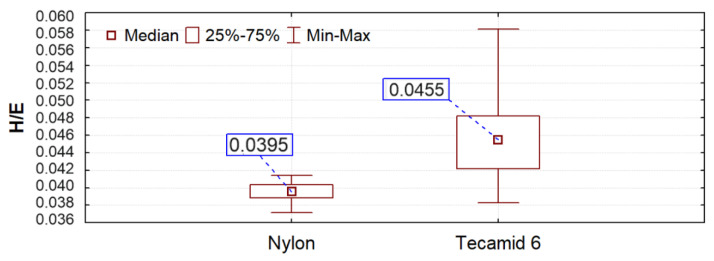
Box plot of H/E ratio calculation results.

**Table 1 materials-16-00573-t001:** Properties of the polyamides.

Property	TECAMID 6	Nylon (Recycled)
Tensile strength (MPa)	79	74
Elongation at field point (%)	4	5
Thermal expansion coefficient (10^−6^/°C)	120	80
Thermal conductivity (W/mK)	0.37	0.23
Glass transition temperature (°C)	45	-
Melting temperature (°C)	221	220
Water absorption (%)	0.3/0.6 (24 h/96 h (23 °C))	1.28 (23 °C 60% RH)

**Table 2 materials-16-00573-t002:** Sliding friction test parameters.

No.	Size	Value
1	Friction angle	360°
2	Linear speed	12.6 cm/s
3	Load	5 N
4	Counter-specimen	ball Ø 6 mm, 100Cr6 material
5	Number of friction cycles	23,830
6	Working fluid (wet sliding)	Distilled water
7	Temperature	22 °C (dry sliding), 50 °C (wet sliding)

**Table 3 materials-16-00573-t003:** Roughness values of tested polymers.

Material/Specimen	$R_{a}\;[\mu m]$	$R_{q}\;[\mu m]$	$R_{v}\;[\mu m]$	$R_{p}\;[\mu m]$	$R_{t}\;[\mu m]$	$R_{z}\;[\mu m]$
Nylon/1	0.32216	0.46414	−2.52253	1.56901	4.09154	2.77806
Nylon/2	0.23711	0.39103	−3.19815	2.36663	5.56478	2.44921
Nylon/3	0.35066	0.44462	−0.94886	1.45052	2.39938	1.71472
Nylon [mean]	0.30331	0.43326	−2.22318	1.79539	4.01857	2.31399
Tecamid 6/1	0.27247	0.35575	−0.91269	1.36947	2.28217	1.15414
Tecamid 6/2	0.22901	0.28245	−0.70058	0.93218	1.63276	1.14156
Tecamid 6/3	0.16228	0.22034	−0.60385	1.20498	1.80884	1.13537
Teceamid [mean]	0.22125	0.28618	−0.73904	1.16888	1.90792	1.14369

**Table 4 materials-16-00573-t004:** Values of the friction coefficient of plastics.

Material (Specimen)	Friction Coefficient µ				
	Initial Value	Minimal Value	Maximum Value	Mean Value	Standard Deviation
Dry sliding friction					
Nylon (1)	0.059	0.059	0.276	0.216	0.052
Nylon (2)	0.018	0.018	0.276	0.242	0.029
Nylon (3)	0.001	0.001	0.280	0.256	0.020
Nylon [mean]	0.026	0.026	0.277	0.238	0.034
Tecamid (1)	0.043	0.043	0.470	0.266	0.107
Tecamid (2)	0.001	0.001	0.593	0.302	0.167
Tecamid (3)	0.001	0.001	0.123	0.108	0.004
Tecamid [mean]	0.015	0.015	0.395	0.225	0.093
Wet sliding friction (50 °C)					
Nylon (1)	0.120	0.076	0.237	0.152	0.016
Nylon (2)	0.119	0.085	0.188	0.135	0.024
Nylon (3)	0.085	0.019	0.209	0.121	0.062
Nylon [mean]	0.108	0.060	0.211	0.136	0.034
Tecamid (1)	0.116	0.095	0.365	0.295	0.045
Tecamid (2)	0.143	0.142	0.154	0.149	0.003
Tecamid (3)	0.093	0.077	0.114	0.097	0.009
Tecamid [mean]	0.105	0.211	0.180	0.019	0.105

**Table 5 materials-16-00573-t005:** Volumetric wear on the surface of the specimens.

Material (Specimen)	V [mm^3^] −10^−3^	
	Dry	Wet 50 °C
Tecamid 6 (1)	14.99	59.15
Tecamid 6 (2)	15.50	90.87
Tecamid 6 (3)	15.25	75.65
Nylon (1)	7.18	41.25
Nylon (2)	8.30	16.69
Nylon (3)	9.35	24.99

**Table 6 materials-16-00573-t006:** Archard’s wear rate and wear factor depending on material specimens.

Material	Tecamid 6						Nylon					
Specimen No.	1		2		3		1		2		3	
	dry	wet	dry	wet	dry	wet	dry	wet	dry	wet	dry	wet
Wear rate *k* [mm^3^/Nm]	4.997 × 10^−6^	1.97 × 10^−6^	5.168 × 10^−6^	30.3 × 10^−6^	5.083 × 10^−6^	25.2 × 10^−6^	2.395 × 10^−6^	13.8 × 10^−6^	2.767 × 10^−6^	5.56 × 10^−6^	3.117 × 10^−6^	8.33 × 10^−6^
Archard’s factor *K*	0.537 × 10^−6^	2.12 × 10^−6^	0.555 × 10^−6^	3.26 × 10^−6^	0.546× 10^−7^	2.71 × 10^−6^	0.288 × 10^−6^	1.65 × 10^−6^	0.333 × 10^−6^	0.669 × 10^−6^	0.375 × 10^−6^	1.00 × 10^−6^

## Data Availability

The data presented in this study are available on request from the corresponding author.

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
