# Peer review of "Comparative Evaluation of the Tribological Properties of Polymer Materials with Similar Shore Hardness Working in Metal–Polymer Friction Systems"

_materials, 2023, doi:10.3390/ma16020573_

Round 1

Reviewer 1 Report

The article reports the tribological properties of polymeric materials which is really scarce in the literature. The authors demonstrated a large number of achieved results that should be published for new research to be initiated. However, for this article to be published, a major review must be carried out.

1. At the beginning of the introduction, the concepts of thermoset and thermoplastic were poorly explained.

2. The text does not show line numbers, so it is difficult to report errors.

3. At the end of the first paragraph, the terms “sustainable development”, “cheaper than many other commercially available” and “being studied is its hardness”: who said that? What were your references?

4. In the article, there is no table 3.

5. During the text, figures 4, 7, 8 and 11 were not discussed.

6. The topic numbers are wrong.

7. In table 2, 4, 5 and 6, for each one (Nylon and Tecamid 6) the average of the 3 values must be presented.

8. In table 6, for better clarification of the results, I suggest putting all values with E-06.

9. On page 17 it was written “such tests”. What were those tests?

10. In figure 11, how were the values of “E” and “H” obtained?

11. In Conclusion 1, no investigation was made of the use of polymers in this article. Conclusions are exclusive to inform the results. In addition, information from the literature should be discussed in the introduction, where the authors should defend the reason for the research.

12. In conclusion 2, the friction coefficients of the two materials were very similar, mainly in a dry environment. How to confirm this self-lubrication if there is not even a SEM image?

13. In conclusion 3, how did the authors speak of "Macrodeformation

Author Response

List of detailed Point-to-Point changes in the manuscript:

Comparative evaluation of the tribological properties of polymer materials with similar Shore hardness working in metal-polymer friction systems (Materials-2103529)

Dear Reviewer,

Thank you very much for your attention and the evaluation and comments on our paper. Those comments are all apt, valuable and very helpful for revising and improving our paper, as well as the important guiding significance to our researches. We have revised the manuscript according to your kind advices and detailed suggestions. Enclosed please find the detailed responses to all your comments, questions and recommendations.

Remark 1:

At the beginning of the introduction, the concepts of thermoset and thermoplastic were poorly explained.

Answer:

The introduction has been supplemented.

Remark 2:

The text does not show line numbers, so it is difficult to report errors.

Answer:

Line numbering has been added.

Remark 3:

At the end of the first paragraph, the terms “sustainable development”, “cheaper than many other commercially available” and “being studied is its hardness”: who said that? What were your references?

Answer:

The introduction text was modified due to the reviewer's comment.

Remark 4:

In the article, there is no table 3.

Answer:

Table numbering has been corrected.

Remark 5:

During the text, figures 4, 7, 8 and 11 were not discussed

Answer:

The text has been supplemented.

Remark 6:

The topic numbers are wrong.

Answer:

The numbering of chapter titles has been corrected.

Remark 7:

In table 2, 4, 5 and 6, for each one (Nylon and Tecamid 6) the average of the 3 values must be presented.

Answer:

The tables have been supplemented with the average value.

Remark 8:

In table 6, for better clarification of the results, I suggest putting all values with E-06.

Answer:

The results in the table have been corrected as suggested.

Remark 9:

On page 17 it was written “such tests”. What were those tests?

Answer:

The sentence has been edited to make it more understandable.

Remark 10:

In figure 11, how were the values of “E” and “H” obtained?

Answer:

References to sources according to which "H" and "E" are determined are given in chapter four and marked in yellow. These values were determined in the test described in section 2.4.

Remark 11:

In Conclusion 1, no investigation was made of the use of polymers in this article. Conclusions are exclusive to inform the results. In addition, information from the literature should be discussed in the introduction, where the authors should defend the reason for the research.

Answer:

The aim of the research and the research problem are explained in the introduction to the article. The first three sentences in the application refer to the possible applications of the tested materials. The tests were carried out for a specific case of the metal-polymer kinematic junction with concentrated contact. In addition, the tests included the impact of the operating hydro-thermal factor. What has a direct application meaning. Two items referred to in the application are to indicate the importance of the undertaken problem of selection of materials based on hardness. Manufacturers do not provide designers with information on tribological properties or those that correlate well with tribological wear of the tested materials. Information only on hardness is insufficient to assess the suitability of the material in the kinematic node in question.

Remark 12:

In conclusion 2, the friction coefficients of the two materials were very similar, mainly in a dry environment. How to confirm this self-lubrication if there is not even a SEM image?

Answer:

We agree with the reviewer. In the light of the presented research results, self-lubrication can only be assumed. The application has been modified. Under macroscopic and organoleptic examination, the surface of the Nylon samples appears to be more "oily" than that of the Tecamid 6 material. Also, the friction surface of the countersamples of the bearing steel balls for the Nylon samples appears to be waxed to a greater extent than the countersamples of the material. Tecamid 6. The coating appears more solid with these countersamples. But this cannot be considered a scientific study. SEM research is an additional separate research work that cannot be done at present. It is possible that the mechanisms of damage in the friction path of polymer samples and "transfer layers" on the friction surface of the balls could be presented. Unfortunately, the available SEM Phenom Pro microscope is damaged (damaged vacuum pump). Even if it was not damaged, in the case of samples made of non-conductive materials, I am not sure whether self-lubrication could be unequivocally demonstrated. We also do not have the funds to commission such tests to another laboratory. It is known that small differences in the coefficient of friction translate into higher differences in wear. In the results contained in this work, the differences in the coefficients of friction are not directly proportional to the measured volumetric wear. It is probably the stability of friction and lower wear of the Nylon material that determine the sliding properties of the tested materials. However, the reviewer's suggestion is valid. In subsequent works, the authors will try to undertake SEM microscopic examinations.

Remark 13:

In conclusion 3, how did the authors speak of "Macrodeformation

Answer:

Exactly. Our task was to explain the possible causes of the tribological behavior of the tested materials. According to the literature [48], macrodeformation is related to the deformation of the material due to the displacement of the contact point on the surface of the polymer element. Viscoelastic materials with deformation hysteresis, also PA, are susceptible to macrodeformation. Macrodeformation occurs especially in cases of cooperation of materials with significantly different hardness, e.g. metal - polymer. The macrodeformation is also related to the range of elastic deformation, which is higher in the case of Tecamid. This is indirectly indicated by the h(eta)it ratio, which is approx. 20% higher. Only indirectly, because it is not a value determined under the conditions of cyclic dynamic load. According to [48], the macro-deformation component of the resistance to motion depends on other factors, including thermal properties, which vary in the case of the tested materials. The thermal expansion of Tecamid 6 is several dozen percent higher. It can be assumed that it will be the highest in the friction contact area. The Tecamid 6 material seems to be more deformable in the case of cyclic concentrated dynamic loads, after exceeding a certain number of friction cycles.

Reviewer 2 Report

This study presents comparative tests of mechanical properties and wear resistance of two commercial polymer materials. Before consideration of acceptance, some comments below are suggested to be addressed.

Q1. Please check “5002,000 “ at section of Surface roughness tests. It should be “500 to 2000”?

Q2. For nanoindentation test, how far apart were each indent spaced? Is there any holding time when the load reached a maximum force of 1 N?

Q3. For formula 4, geometric property of Vickers indentation is not “24.54”. Please check that.

Q4. For formula 5, please define i and Ei.

Q5. Page 9, the name, “N,” for mean value is duplicate.

Q6. There is no description for Figure 7 and 8 in manuscript. Please improve that.

Q7. For friction coefficient tests, authors use number of cycles to discuss friction coefficient tests. However, friction path (or wear distance) is much useful than number of cycles. Could Authors revise cycles to distance?

Q8. In the last paragraph at section of analysis and discussion, there is no Figure 12. It should be figure 11. Please check that.

Q9. In page 17, number of the formula for wear index, k, is incorrect. Please revise it.

Author Response

List of detailed point-to-point changes in the manuscript:

Comparative evaluation of the tribological properties of polymer materials with similar Shore hardness working in metal-polymer friction systems (Materials-2103529)

Dear Reviewer,

Thank you very much for your attention and evaluation and comments on our paper. Those comments are all apt, valuable and very helpful for revising and improving our paper, as well as the important guiding significance to our researches. We have revised the manuscript according to your kind advice and detailed suggestions. Enclosed, you will find the detailed responses to all your comments, questions and recommendations.

Remark 1:

Q1. Please check “500¸2,000 “ at section of Surface roughness tests. It should be '500 to 2000'?

Answer:

The text has been corrected; the suggested change has been made.

Remark 2:

Q2. For the nanoindentation test, how far apart were each indent spaced? Is there any holding time when the load reached a maximum force of 1 N?

Answer:

The holding time was 20 s. 12 measurements were made on each sample. According to the scheme shown below.

(see attachment for pictures) 

Remark 3:

Q3. For formula 4, the geometric property of Vickers indentation is not '24.54'. Please check that.

Answer:

The text has been corrected; the suggested change has been made.

Remark 4:

Q4. For formula 5, please define ni and Ei.

Answer:

The text has been corrected; the suggested change has been made.

Remark 5:

Q5. On page 9, the name 'N' for mean value is duplicate.

Answer:

The text has been corrected; the duplicate mark has been removed.

Remark 6:

Q6. There is no description for Figures 7 and 8 in the manuscript. Please improve that.

Answer:

The text has been corrected; the suggested change has been made.

Remark 7:

Q7. For friction coefficient tests, authors use number of cycles to discuss friction coefficient tests. However, friction path (or wear distance) is much more useful than number of cycles. Could Authors revise cycles to distance?

Answer:

The authors believe that the number of friction cycles is more useful in assessing wear than the friction path. When the friction path is circular, the number of full laps is more important than the friction path. Because in practice, several tests with different friction radii are performed on the same sample. In the dry friction test, the friction distance was 600 m, in hydrothermal conditions the friction path was 898 m.

Remark 8:

Q8. In the last paragraph in the section of analysis and discussion, there is no Figure 12. It should be Figure 11. Please check that.

Answer:

The mistake has been corrected, and the text has changed.

Remark 9:

Q9. On page 17, the number of the formula for wear index, k, is incorrect. Please revise it.

Answer:

The numbering of mathematical formulas has been corrected throughout the text.

Round 2

Reviewer 1 Report

I have read through the revised manuscript and, focusing on the changes made by the authors in response to the many comments and suggestions I offered in my original review. From my perspective, the authors have done a good job of revising the manuscript accordingly (and also responding to the other reviewers comments/questions). I judge the paper to suitable for publication in its current form and think it will become an interesting and valuable addition to the literature.

Reviewer 2 Report

Authors have addressed all my comments. Method and discussion have been significantly improved. Therefore, I recommend accepting present version